# Systematic review on the compliance of WHO guidelines in the management of patients with advanced HIV disease in Africa: The case of cryptococcal antigen screening

**Zuhura Mbwana Ally**[1]\*, **Jackline Vicent Mbishi**[2], **Mariam Salim Mbwana**[3], **Hafidha Mhando Bakari**[4], **Swalehe Mustafa Salim**[5], **Zarin Nudar Rodoshi**[6], **Muhidin Ibrahim Hundisa**[7], **Rebecca Mesfin Sileshi**[8], **Biruk Demisse Ayalew**[8], **Rahma Musoke**[9], **Lynn Moshi**[10], **Yousef Elias Fakhoury**[11], **Haji Mbwana Ally**[12], **Habib Omari Ramadhani**[12]

1 District Hospital, Tanga, Tanzania, 2 Muhimbili University of Health and Allied Sciences, Dar es Salaam, Tanzania, 3 Primary Health Care Institute, Iringa, Tanzania, 4 University of Dar es Salaam, Dar es Salaam, Tanzania, 5 Canada Youth Group, Dar es Salaam, Tanzania, 6 Mymensingh Medical College & Hospital, Mymensingh, Bangladesh, 7 Ministry of Health, Addis Ababa, Ethiopia, 8 St. Paul's Hospital Millennium Medical College, Addis Ababa, Ethiopia, 9 Water Mission, Dar es Salaam, Tanzania, 10 Aga Khan Hospital, Dar es Salaam, Tanzania, 11 Al Hussain New Salt Hospital, As-Salt, Jordan, 12 Kilimanjaro Christian Medical Center, Kilimanjaro, Tanzania

\* zuhuraally86@gmail.com

**Data Availability Statement:** All relevant data are within the manuscript and its Supporting

## Abstract

### Background

The World Health Organization (WHO) recommended cryptococcal antigen (CrAg) screening for people presenting with advanced HIV disease (AHD) and for those with positive CrAg without evidence of meningitis to initiate preemptive antifungal medication. Data on the implementation of WHO recommendations regarding CrAg screening is limited. We estimated pooled prevalence of CrAg screening uptake, cryptococcal antigenemia, lumbar puncture, cryptococcal meningitis and initiation of preemptive antifungal medication from available eligible published studies conducted in Africa.

### Methods

PubMed, Cochrane Library and Embase were searched for articles published between January 2011 and December 2023. CrAg uptake was defined as percentage of eligible people (CD4 $\leq$ 200 cells/mm$^3$ or WHO stage III/IV) who received cryptococcal antigen testing. Stratified analysis to compare uptake and cryptococcal antigenemia between studies that involved multiple vs single sites was performed. Using random effects models, we computed the pooled estimate of CrAg screening uptake, cryptococcal antigenemia, lumbar puncture, cryptococcal meningitis, preemptive antifungals treatment and 95% confidence intervals (CIs).

### Results

Ten studies with 18,820 individuals with AHD were analyzed. Overall, the pooled estimate of CrAg screening uptake was 57.1% (95% CI: 41.4–72.7). CrAg screening uptake was

Information files. Extracted dataset has been provided.

**Funding:** The author(s) received no specific funding for this work.

**Competing interests:** The authors have declared that no competing interests exist.

significantly lower among studies that involved multiple sites compared to those that involved single site, (47.3% vs 73.3%; p<0.001). Overall, the pooled prevalence of cryptococcal antigenemia was 9.6% (95% CI:6.4–12.9). Cryptococcal antigenemia was significantly lower among studies that involved multiple sites compared to those that involved single site, (9.1% vs 10.4%; p<0.001). Among those who tested positive for CrAg, 84.6% (95% CI: 54.1–99.0) received preemptive antifungal treatment, though nearly 25% did not undergo lumbar puncture, highlighting gaps in diagnostic follow-up. Six studies evaluated CrAg positive patients with lumbar puncture and the overall prevalence of lumbar puncture was 74.9% (48.0–94.8). The overall prevalence of cryptococcal meningitis was 58.1% (46.6–69.6).

## Conclusions

Not screening for CrAg among people with AHD and failure to initiate antifungal medications among eligible patients with cryptococcal antigenemia presents a significant missed opportunity. Emphasis on improving CrAg screening is critical given its proven cost-effective benefits.

## Introduction

Cryptococcal meningitis is the second most common cause of HIV related deaths after tuberculosis [1]. Estimates from the Joint UN Programme on HIV and AIDS (2019 to 2020) and population-based HIV impact assessment surveys showed that, globally, cryptococcal disease account for 19% of AIDS-related mortality [2]. Annually, 152,000 cases of cryptococcal meningitis are estimated to occur among people living with HIV (PLWH) with CD4 count <200cells/mm$^3$ globally [2]. Low- and mid-income countries (LMICs) are the most hit by cryptococcal meningitis even during this period of an increased access to antiretroviral therapy (ART). Of these estimated annual incident cases of cryptococcal meningitis, 73% occurred in Africa and 19% in Asia and Pacific [3]. ART-naïve people living with HIV who present with advanced HIV disease (AHD), as defined by the World Health Organization (WHO) (CD4 count ≤ 200 cells/mm$^3$ or WHO stage III/IV), are at significantly higher risk of cryptococcal meningitis [4, 5].

Screening for cryptococcal meningitis is a highly cost-effective intervention [6–8], with the potential to avert up to 43% of deaths related to this infection among PLWH presenting with AHD [7]. Provision of antifungal medications such as fluconazole among PLWH who did not have cryptococcal meningitis but screened positive for cryptococcal antigen CrAg prevents the development of cryptococcal meningitis and deaths [9]. One meta-analysis that involved nine studies of which six (56%) were from Africa showed that, the incidence of cryptococcal meningitis among CrAg positive patients (CD4 count < 200 cells/mm$^3$ and without evidence of cryptococcal meningitis) and received antifungals was 3% compared to 5% among those who did not receive antifungals, equivalent to 40% reduction in the development of cryptococcal meningitis [10]. Another systematic review that involved 31 studies of which 21 (68%) were from Africa showed that among those with CD4 count < 100 cells/mm$^3$, the incidence of cryptococcal meningitis was 21.4% without preemptive fluconazole compared to 5.7% with preemptive fluconazole [11]. Due to these notable benefits, the WHO recommended regular screening for cryptococcal disease among PLWH presenting with AHD and for those with detectable CrAg but without evidence of cryptococcal meningitis to receive preemptive antifungal medications [12].

Several studies have evaluated the prevalence of positive CrAg among ART naïve PLWH presenting with AHD [13–15] and a few also explored CrAg positivity among ART experienced PLWH either with unsuppressed viral load [16] or with suppressed and unsuppressed viral load [17]. While understanding the magnitude of CrAg positivity is critical, there is limited data on the uptake of CrAg implementation screening following WHO recommendations. CrAg uptake reported in Uganda was 19% in 2019 [7] and 95% in 2021 [16] among ART naïve patients. In 2022, data from a single site in Malawi also reported CrAg uptake of 89% [18]. The variations in the sources of data used (single site vs multi-site; levels of health care facilities) could attribute to the differences in the reported CrAg screening uptake in these settings. Moreover, supplies stock out, dysfunctional CD4 count instruments, lack of training and unawareness of the WHO guidelines are some of the identified reasons for the low uptake of CrAg screening as reported by prior studies [19–21]. Among patients who tested positive for CrAg without evidence of cryptococcal meningitis, 54% to 100% are documented to have initiated preemptive antifungal medications [22, 23]. In this systematic review and meta-analysis, we estimated pooled prevalence of CrAg screening uptake, cryptococcal antigenemia, lumbar puncture, cryptococcal meningitis and initiation of preemptive antifungal medication in Africa.

## Methods

### Study protocol and registration

This systematic review has been registered in the International Prospective Registry of Systematic Review with registration number CRD42023494535. This systematic review and meta-analysis was reported according to the Preferred Reporting Items for Systematic Reviews and Meta-Analyses (PRISMA) guidelines (S1 Table) [24].

### Ethical approval

Because this was a systematic review of published manuscripts, ethical approval was not sought.

### Search strategy

PubMed, Cochrane CENTRAL, Embase and clinicaltrials.gov were searched by a medical librarian (EL) for the articles published between January 2011 and December 2023. Search terms were used to capture information on uptake of cryptococcal meningitis screening among PLWH who presented with advanced HIV disease in Africa. A full search strategy can be found in S2 Table. The search was restricted to papers written in English. Search results were uploaded to Covidence Systematic Review Software (Melbourne, Australia) for deduplication and screening.

### Eligibility criteria

The eligibility criteria were structured using the PICOS (ie, population, intervention, comparison, outcome, and study type) framework [25] as follows: People with advanced HIV disease (CD4 $\leq$ 200 cells/mm$^3$ or WHO stage III/IV) as the population. Stratified analysis was conducted to understand differences/similarity of outcome between pooled estimates resulting from studies conducted from multiple sites vs single sites. Multiple study sites was then regarded as the intervention/exposed groups; single study sites as comparison groups; CrAg screening uptake, cryptococcal antigenemia, and preemptive antifungals initiations, lumbar puncture and cryptococcal meningitis as outcome; and observational studies as study design.

## Inclusion and exclusion criteria

Observational studies conducted in Africa reported CrAg screening uptake among PLWH with advanced HIV disease and/or proportions of eligible persons who received preemptive antifungal medications and written in English were eligible for inclusion. People eligible for CrAg screening included those with advanced HIV disease as defined by WHO as those with $CD4 \leq 200$ cells/mm$^3$ or WHO stage III/IV. We included studies that reported CrAg uptake from 2011 when WHO guidelines requiring screening for CrAg was launched. We excluded studies that did not report the uptake of CrAg screening or reported CrAg uptake prior to the implementation of WHO guidelines.

## Study selection and data extraction

The manuscripts searched from outlined databases were managed by Covidence software, from which the final list of manuscripts was deduplicated. Two pairs of review authors (ZMA/HRO and ZNR/BDA) independently completed the study selection for inclusion in the appraisal process. Disagreement between two independent pairs of reviewers for the inclusion of the manuscripts was handled by the third pair of reviewers (SMS/MIH). Using a pre specified Excel spread sheet template, the two pairs of review authors (ZMA/HRO and ZNR/BDA) independently extracted the following data elements from the included studies: authors, year of publication, the country in which the study was conducted, study design, study period, status of ART at the time of screening (ART naïve vs both ART naïve and experienced), study sites (single vs multi sites), CrAg eligibility criteria ($CD4 \leq 100$ or $CD4 \leq 200$), CrAg screening assessment (Provider initiated or lab reflex initiated), number of people eligible for CrAg screening, number of people who were screened for CrAg, number of people who were positive for CrAg, number of people who received preemptive antifungal medications, and study sample size. Number of people who had lumbar puncture and number of people with cryptococcal meningitis was also extracted.

## Risk bias assessment

The National Institute for Health (NIH) tool was used to assess the quality studies [26]. Reliability and validity of the measurement tools, participation rate, source of study participant recruitment, justification of sample size or power calculations and appropriateness of statistical analysis were evaluated for all studies that met inclusion criteria. Finally, percentage scores reflecting the degree of bias from the studies were computed based on the stated parameters.

## The overall quality of systematic review

GRADE (Grading of Recommendations, Assessment, Development, and Evaluations) framework was used to assess the overall quality of systematic review. GRADE was used to rate the body of evidence at the outcome level and has four levels of quality of evidence: very low, low, moderate, and high. Each of the five domains risk of bias, consistency, precision, directness and publication bias were assessed. Risk of bias was assessed using the NIH tool. Criteria for judgement the presence of bias was used as previously described [27]. Consistency was assessed by observing similarity of point estimates, extent of overlap of confidence intervals, and statistical tests of heterogeneity [28]. Precision was assessed using pooled effect size, 95% CI and the optimal information size (OIS) which refers to sample size required to detect the difference for the adequately powered study [29]. For this assessment, we considered effect size of the effect of assessing CrAg uptake using studies conducted from multiple sites vs single sites. The OIS to detect 10% difference in CrAg uptake among studies conducted from

multiple sites (assumed to be 65%) compared to those conducted from single site (assumed to be 75%) was 658 participants. Directness was assessed by looking if intervention and outcome of interest were measured from the intended population or surrogate indicators were used [30]. Publication bias was assessed using Egger test from funnel plots assessment as previously described [31].

### Definition of variables

**Exposure and outcome.** The main outcome of interest is the uptake of CrAg screening defined as percentage of eligible PLWH (CD4 $\leq$ 100 cells/mm$^3$ or $\leq$ 200 cells/mm$^3$ or WHO stage III/IV) who were screened for CrAg. Other secondary outcomes included prevalence of positive CrAg and preemptive antifungal medications initiations. Furthermore, among those who screened positive for CrAg in serum, prevalence of lumbar puncture and cryptococcal meningitis was also evaluated. Uptake of CrAg screening and CrAg positivity were further stratified by the study sites (multi-sites vs single-site). The differentiation between studies conducted in single-site (e.g., tertiary hospitals) vs multi-site (e.g., national and district healthcare settings) was performed as these variations could influence both CrAg uptake and subsequent treatment initiation.

**Data synthesis and analysis.** Meta regression for proportions using standard random effects models were used to compute pooled prevalence of CrAg uptake, CrAg positivity, receipt of preemptive antifungal medications, lumbar puncture and cryptococcal meningitis. The variance of proportions were stabilized using Freeman-Tukey double arcsine transformation before pooling [32, 33]. Subgroup analysis on the pooled prevalences of these outcomes were performed based on the study site (multi-sites vs single site). Two sample tests for proportion were used to compare prevalences of CrAg uptake and positivity of the stratified analysis. We evaluated heterogeneity across studies using the $I^2$ statistic, which represents the percentage of total variability due to between-study heterogeneity. The $I^2$ statistic was categorized at values of 25%, 50% and 75% to represent low, moderate, and high extent of heterogeneity respectively as previously described [34]. The publication bias was assessed using the Egger regression asymmetry test. For both heterogeneity and publication test, a p value < 0.05 indicated the presence of heterogeneity and publication bias respectively. To explore the source of heterogeneity, an influential analysis using the leave-one-out method was performed. Studies with missing information, such as those that reported proportions of outcomes without actual numerators and/or denominators, were excluded from the analysis. All statistical tests were performed using STATA version 17 (Stata Corporation, College Station, Texas, USA).

## Results

A total of 2,688 publications were retrieved through searches and 846 were duplicates and removed, leaving 1,842 publications to have their titles and abstracts screened and 1,827 were excluded as they did not address the intended study matter. The remaining 15 received a full review and 10 were eligible for final analysis (Fig 1). Dataset for all 10 studies can be found at S3 Table. A summary of all identified studies, those excluded and reasons for exclusion can be found at S4 Table.

### Study selection

**Characteristics of studies included.** Study articles included participants from multiple sites (n = 6) [16, 19, 21, 22, 35, 36] and participants from single sites (n = 4) [14, 18, 23, 37]. Of the ten studies included, five included ART naïve and other five included both ART naïve and ART experienced participants. Sample size for studies ranged from 109 to 8,147. A total of

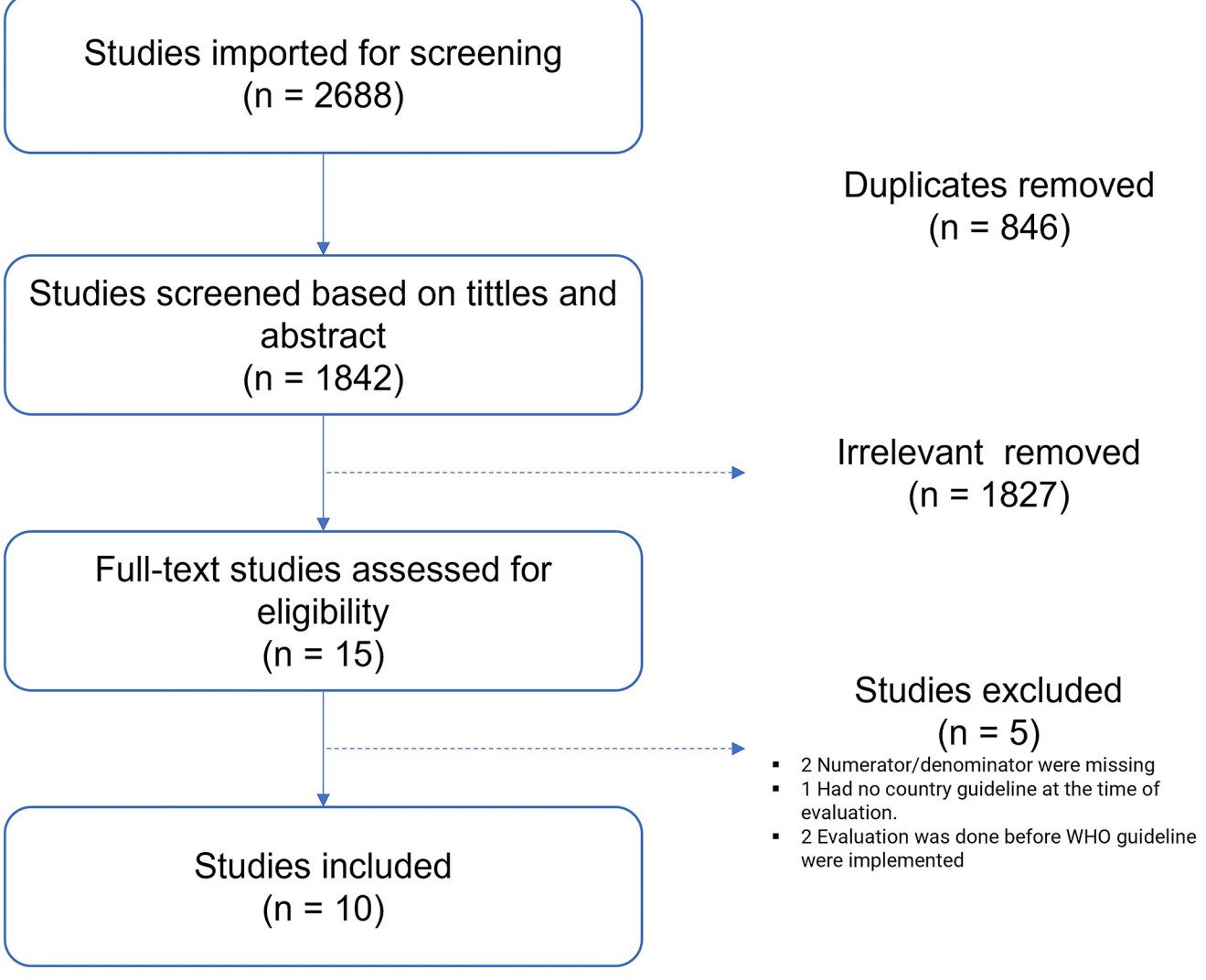

**Fig 1. PRISMA flow diagram of the included studies for meta-analysis of on the compliance of WHO guidelines in the management of patients with advanced HIV disease in Africa.**

18,820 PLWH who had AHD were included in this analysis (Table 1). We included studies conducted from nine countries in Africa.

**CrAG assessments and sources of data.** At the time of CrAg assessments, all country guidelines recommended screening among patients with advanced HIV disease. CrAg assessments was done using routinely collected data from health care facilities. Most studies 8/10 evaluated CrAg screening from data collected on provider-initiated scheme, one study evaluated CrAg screening using data collected on provider-initiated scheme and laboratory reflex and one study evaluated CrAg screen on laboratory reflex only (Table 2).

## Risk of bias assessment

All studies included demonstrated high scores with median scores of 85% [interquartile range (71–89)], indicating very low risk of bias. Failure to provide justification of the sample size and/or power calculations was the most common methodological weakness. The overall risk of bias assessment can be found at S5 Table.

Table 1. Summary of characteristics of the included studies.

| Author | Study period | Country | CrAg criteria | ART status at the time of screening | Eligible for CrAg screening | CrAg screening uptake | Prevalence of positive CrAg | Preemptive antifungal initiations | Quality scores % |
|---|---|---|---|---|---|---|---|---|---|
| Baluku et al, 2021 [16] | 2015–2018 | Uganda | CD4 < 100 | Both | 8,147 | 21.1 | 12.2 | NR | 86 |
| Enock et al, 2022 [19] | 2015–2017 | Uganda | CD4 ≤ 100 | Naïve | 359 | 71.0 | 22.0 | 83.9 | 92 |
| Tiam et al, 2023 [21] | 2018–2019 | Lesotho | CD4 < 200 | Naïve | 109 | 35.8 | 12.8 | NR | 71 |
| Blankley et al, 2019 [14] | 2015–2016 | Zimbabwe | CD4 < 100 | Naïve | 377 | 83.0 | 8.0 | NR | 85 |
| Vallabhaneni et al, 2016 [22] | 2012–2013 | South Africa | CD4 < 100 | Naïve | 4,395 | 26.6 | 2.1 | 54.2 | 71 |
| Drain et al, 2021 [23] | 2013–2019 | South Africa | CD4 ≤ 200 | Naïve | 908 | 33.1 | 9.3 | 100.0 | 71 |
| Heller et al, 2022 [18] | 2017–2020 | Malawi | CD4 < 200 | Both | 475 | 89.1 | 13.9 | NR | 92 |
| Sing'oei et al, 2017 [35] | 2013–2017 | Multi country* | CD4 < 200 | Both | 494 | 80.4 | 3.8 | NR | 86 |
| Kanyama et al,2022 [37] | 2016–2017 | Malawi | CD4 ≤ 200 | Both | 221 | 81.0 | 10.6 | NR | 89 |
| Hurt et al, 2021 [36] | 2014–2016 | Botswana | CD4 ≤ 100 | Both | 3,335 | 49.3 | 6.3 | NR | 85 |

Abbreviations: CrAg indicates cryptococcal antigen; NR indicates not reported; * indicates Uganda, Kenya, Tanzania, and Nigeria.

## CrAg screening uptake

Across all studies, the pooled estimate of CrAg screening uptake was 57.1% (95% CI 41.4–72.7) (Fig 2), with significant variations observed between studies conducted from multiple vs single sites. Table 3 describes results of stratified analysis on the prevalence of CrAg screening uptake. Pooled prevalence of CrAg screening uptake was significantly lower among studies that involved multiple sites 47.3%(95% CI 31.9–63.0) compared to 73.3% (95% CI 40.6–95.9) among studies that involved single site (p < 0.001). Single-site studies, often conducted in tertiary hospitals, generally had better CrAg uptake due to more consistent resources, such as trained staff and reliable access to CD4 count machines. Conversely, multi-site studies, which included smaller or district-level healthcare facilities, reported lower uptake, likely due to resource limitations.

## Assessment of heterogeneity, publication bias, and influential analysis on the CrAg screening retesting uptake

The $I^2$ statistic test of heterogeneity showed a value of 99% (p = 0.0) on the overall CrAg screening uptake, indicating high heterogeneity (Table 3). Egger's test showed no statistically significant small-study effect (z-test = 0.57, p = 0.571), indicating absence of publication bias. Estimates from the leave-one-out forest plot showed a non-significant influence of all studies on the CrAg screening uptake (Fig 3).

## Prevalence of CrAg antigenemia

Six thousand four hundred and forty-three people with AHD had positive CrAg, translating to an overall pooled prevalence of 9.6% (95% CI 6.4–12.9) (Fig 4). The pooled prevalence of

**Table 2. Context descriptions of included studies.**

| | Study setting and population | Number of sites | Country guidelines at the time of screening | CrAg screening assessment | Design and data sources |
|---|---|---|---|---|---|
| Baluku et al, 2021 [16] | All people living with HIV who receive routine HIV care from facilities of different administrative level in eight rural districts. | 104 facilities (1 regional referral hospital, four district hospitals, sixteen health center IVs, seventy-nine health center IIIs, and four health center IIs | ART-naïve people with HIV were required to have CD4 count measurement and CrAg screening if CD4 count is < 100 cells/mm$^3$ | Routine assessment with provider initiated | Review of routinely generated programmatic data. |
| Enock et al, 2022 [19] | All people living with HIV who receive routine HIV care from facilities of different administrative level in five districts (2-Urban and 3-Rural) | Fourteen facilities. The fourteen health facilities (Six health center level three, three health center level four, three general referral hospitals, and two regional referral hospitals) | ART-naïve people with HIV were required to have CD4 count measurement and CrAg screening if CD4 count is ≤ 100 cells/mm$^3$ | Routine assessment with provider initiated | Retrospective review of medical records from CD4 and CrAg registers(standard Uganda Ministry of Health (MoH) tools that are used for documentation and generation of routine performance reports |
| Tiam et al, 2023 [21] | Enrolled 15 years or older people with AHD (CD4 < 200 cells/mm$^3$ or WHO stage 3/4) | Two largest hospitals ART clinics at the Motebang and Berea District. The hospital serves one-third of Lesotho's population. | Same-day serum CrAg screening test for all patients enrolling in care with CD4 count < 200 cells/mm$^3$ | Routine assessment with provider initiated. Job aids developed to improve clinician awareness and facilitate assessment | Prospective evaluation of routinely collected data from ART clinics. Follow up 6 months |
| Blankley et al, 2019 [14] | Enrolled 19 years or older ART naïve people with AHD (CD4 < 200 cells/mm$^3$ or WHO stage 3/4) at a semi-urban polyclinic in Epworth, Zimbabwe. | Single site study at Epworth polyclinic (a nurse led with support from Me´decins Sans Frontières and Ministry of Health doctors | From 2015, recommended CrAg screening for those with CD4 < 100 cells/mm$^3$ | Routine assessment with provider initiated | Retrospective assessment of outcomes and management of patients with advanced HIV disease. |
| Vallabhaneni et al, 2016 [22] | Enrolled 18 years or older with a CD4 count of <100 cells/µL at ART clinics in Western Cape Province. | Multiple ART facilities in Western Cape province | CrAg screening instructing clinicians to offer CrAg screening to ART naïve with a CD4 cell count <100 cells/µl. | Routine assessment with provider initiated | Retrospective evaluation of the CrAg screen to identify the proportion of eligible patients screened for CrAg. Data from, pre-existing provincial and national databases. |
| Drain et al, 2021 [23] | Enrolled 18 years or older people from voluntary counselling and testing site. | Single site at iThembalabantu Clinic | Phase I: September 12$^{th}$ 2013 to June 4$^{th}$, 2015- CrAg screening for with CD4 < 100 cells/mm$^3$ (Provider initiated) Phase II: June 5$^{th}$, 2015, to November 8$^{th}$, 2017- CrAg screening for with CD4 < 100 cells/mm$^3$ (Reflex testing) Phase III: November 9$^{th}$ 2017 to February 28$^{th}$, 2019- the team performed point of care test and CrAg for those with CD4 ≤ 200 cells/mm$^3$ | Mixed assessments Provider initiated and Lab reflex testing | Evaluation of standard of care CrAg testing and positivity to quantify Cryptococcus diagnosis and treatment and to determine the delivery of Cryptococcus guidelines. |
| Heller et al, 2022 [18] | In patient evaluation of ADH management practices at a tertiary hospital in Malawi | Single site at Kamuzu Central Hospital. A tertiary hospital located in Lilongwe. | 2017: Recommended CrAg screening for those with CD4 < 100 cells/mm$^3$ 2020: Recommended CrAg screening for those with CD4 < 200 cells/mm$^3$ | Routine assessment with provider initiated AHD. An HIV counselor, a nurse, and a part time clinical officer staffed the AHD room to facilitate management of patients with AHD. | Evaluation of outcome for patients with AHD using routinely collected data |

(*Continued*)

**Table 2.** (Continued)

| | Study setting and population | Number of sites | Country guidelines at the time of screening | CrAg screening assessment | Design and data sources |
|---|---|---|---|---|---|
| Sing'oei et al, 2017 [35] | CrAG screening for people with HIV whose CD4 < 200 cells/mm$^3$ from multiple facilities in 4 countries | 11 PEPFAR-supported facilities in Uganda, Kenya, Tanzania, and Nigeria. | All 4 countries guidelines recommended CrAg screening for those with CD4 < 200 cells/mm$^3$ | Routine assessment with provider initiated | Retrospective evaluation of routinely collected from health care facilities |
| Kanyama et al,2022 [37] | Enrolled 14 years or older in patients from a tertiary hospital. AHD diagnosis based on CD4 < 200 cells/mm$^3$ | Single site at Kamuzu Central Hospital. A tertiary hospital located in Lilongwe | Between 1 August 2016 and 31 January 2017, CD4 cell count, urine LAM,urine X-pert and CrAg screening services for management of AHD were introduced | Routine assessment with provider initiated | Prospective evaluation of routinely collected data from medical wards among patients with AHD |
| Hurt et al, 2021 [36] | Tested laboratory samples for patients with AHD based on CD4 count < 100 cells/mm$^3$ | Data from 27 ART clinics and one central referral hospital in Botswana | Recommended CrAg screening for those with CD4 ≤ 100 cells/mm$^3$ | An evaluation of Reflex Cryptococcal Antigen Screening | Evaluation of data from the Botswana Harvard HIV Reference Laboratory |

Abbreviations: AHD indicates advanced HIV disease; HIV indicates Human immune deficiency virus; ART indicates antiretroviral therapy

positive CrAg was significantly lower 9.1% (95% CI 4.9–13.3) among studies that involved multiple sites compared to 10.4% (95% CI 7.7–13.1) among studies that involved single sites (p < 0.001) (Table 3).

## Assessment of heterogeneity, publication bias, and influential analysis on the prevalence of cryptococcal antigen

There was high heterogeneity with I$^2$ statistic of 98% (p = 0.0) on the overall prevalence of positive CrAg (Table 3). Egger's test showed statistically non-significant small-study effect (z-test = 1.72, p = 0.085), indicating absence of publication bias. Estimates from the leave-one-out forest plot showed a non-significant influence of all studies on the prevalence of CrAg (Fig 5).

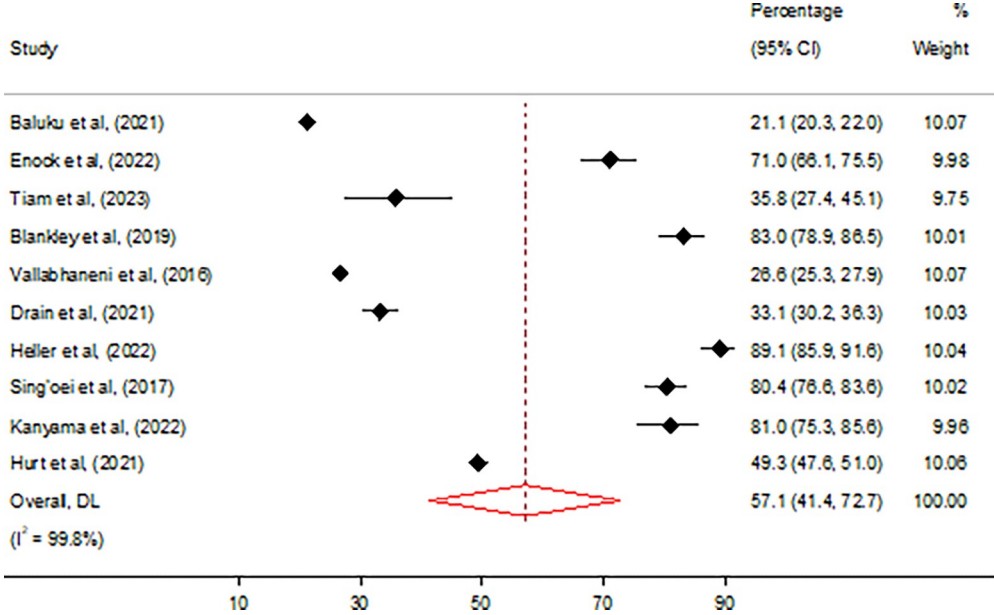

**Fig 2. Overall prevalence of CrAg screening uptake among people living with HIV in Africa.**

**Table 3. Cryptococcal meningitis screening uptake, cryptococcal antigenemia and preemptive antifungal medication initiations among people living with HIV in Africa.**

| Characteristic | # of studies | n | Prevalence (95%CI) | $I^2$% | p-value | Egger's test (z) | p-value |
|---|---|---|---|---|---|---|---|
| CrAg screening uptake | | | | | | | |
| Overall | 10 | 18,820 | 57.1 (41.4–72.7) | 99 | 0.0 | 0.57 | 0.571 |
| Single site [14, 18, 23, 37] | 4 | 1,981 | 73.3 (40.6–95.9) | 99 | 0.0 | 5.98 | < .001 |
| Multi-site [16, 19, 21, 22, 35, 36] | 6 | 16,839 | 47.3 (31.9–63.0) | 99 | 0.0 | 9.41 | < .001 |
| CrAg antigenemia | | | | | | | |
| Overall | 10 | 6,443 | 9.6 (6.4–12.9) | 98 | 0.0 | 1.72 | 0.085 |
| Single site [14, 18, 23, 37] | 4 | 1,216 | 10.4 (7.7–13.1) | 97 | 0.1 | 7.52 | < .001 |
| Multi-site [16, 19, 21, 22, 35, 36] | 6 | 5,227 | 9.1 (4.9–13.3) | 97 | 0.0 | 4.21 | < .001 |
| Preemptive antifungal initiation | | | | | | | |
| Overall [14, 19, 22] | 3 | 88 | 84.6 (54.1–99.0) | 91 | 0.0 | -0.20 | 0.841 |
| Lumbar puncture (LP) | | | | | | | |
| Overall [14, 18, 19, 21, 36, 37] | 6 | 129 | 74.9 (48.0–94.8) | 93 | 0.0 | 1.73 | 0.084 |
| Results of LP | | | | | | | |
| Overall [14, 18, 21, 37] | 4 | | 58.1 (46.6–69.6) | 2.6 | 0.3 | 0.89 | 0.376 |

Abbreviations: n, indicates number; CI, indicates confidence intervals; CrAg indicates cryptococcal antigen

## Prevalence of preemptive antifungal medications initiations

Only three studies reported preemptive antifungal medications initiations, and an overall pooled prevalence was 84.6% (95% CI 54.1–99.0) (Table 3).

## Cryptococcal meningitis evaluation among CrAg positive patients

A total of six studies reported cryptococcal meningitis evaluation following serum CrAg screening [14, 18, 19, 21, 36, 37]. The overall pooled prevalence of lumbar puncture was 74.9%

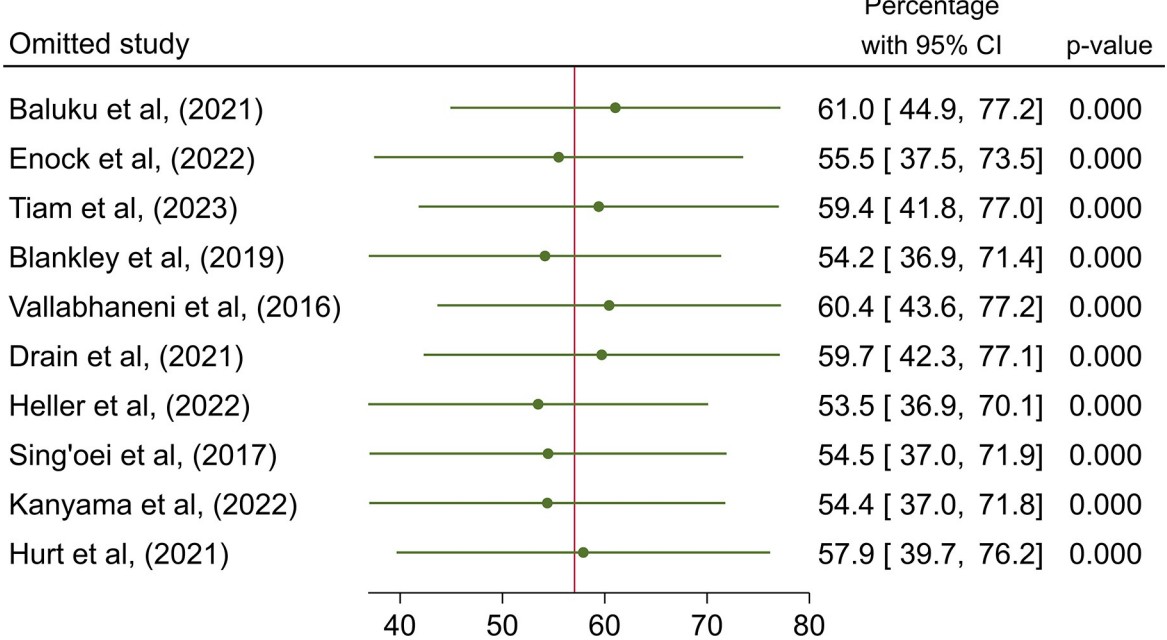

**Fig 3. Influential analysis on the overall prevalence of CrAg screening uptake among people living with HIV in Africa.**

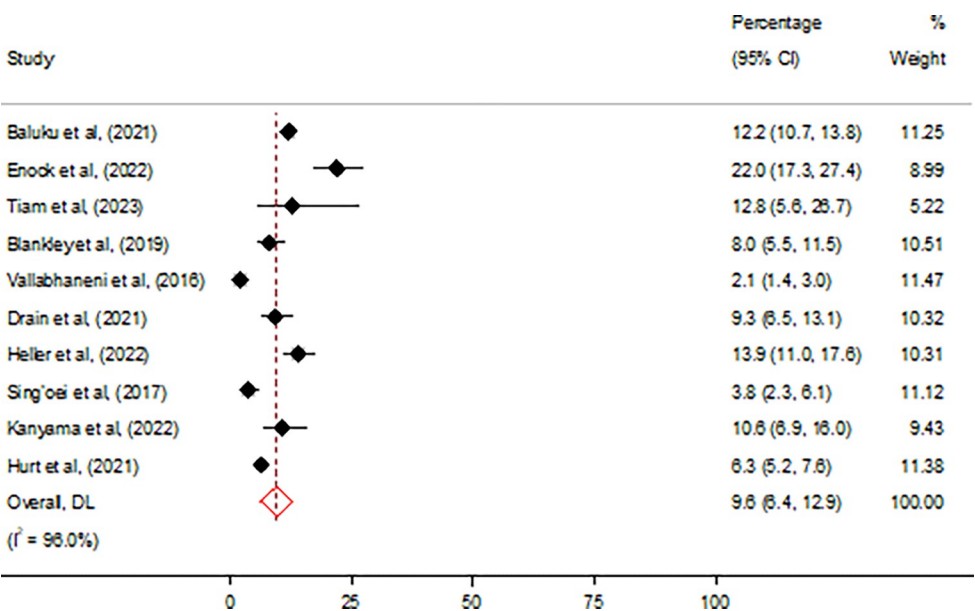

**Fig 4. Overall prevalence of cryptococcal antigenemia among people living with HIV in Africa.**

(48.0–94.8). Of the six studies that conducted lumbar puncture to patients with AHD, four reported lumbar puncture results and the pooled prevalence of cryptococcal meningitis was 58.1% (46.6–69.6).

## Overall quality of systematic review

The overall sample size (n = 18,820) is larger than the estimated OIS (n = 658) and the 95% CI interval for the difference in CrAg uptake between studies conducted between multi vs single

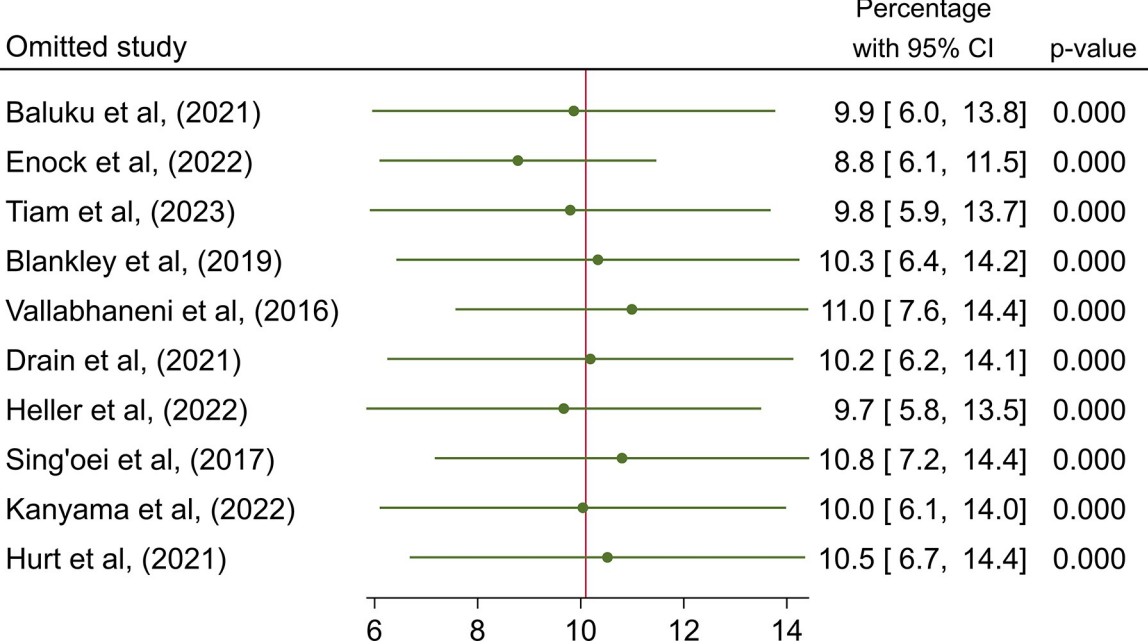

**Fig 5. Influential analysis on the overall prevalence of cryptococcal antigenemia among people living with HIV in Africa.**

sites excluded the null value indicating high level of precision. Five out of ten studies showed CrAg uptake estimates between (71–89) % and the other five showed CrAg uptake estimates between (21–49) % with high degree of heterogeneity indicating low level of consistency. All studies included in this review measured same data from the intended study population, no surrogate markers were used, indicating high level of directness. The median scores for risk of bias assessment was 85% indicating low risk of bias. Overall, Egger test did not show presence of publication bias. When GRADE framework was used to assess the quality of other outcomes (cryptococcal antigenemia, preemptive antifungal medication initiation and lumbar puncture) results were similar to CrAg uptake in that, there was high heterogeneity translating to low consistency, high level of directness and absence of publication bias. Similarly, with large sample size along with the exclusion of null value in the 95% CI interval for the difference in CrAg antigenemia between studies conducted between multi vs single sites showed high level of precision for CrAg antigenemia. Due to the smaller numbers of studies involved in other outcomes, stratified analysis was not performed, and precision was not reported. Overall summary of quality of systematic review using GRADE system framework can be found at S6 Table.

## Discussion

We conducted a systematic review and meta-analysis among PLWH in Africa to evaluate pooled estimates of uptake of CrAg screening, cryptococcal antigenemia, lumbar puncture, cryptococcal meningitis and preemptive initiation of antifungal medications. The overall pooled estimates of uptake of CrAg screening, cryptococcal antigenemia and preemptive initiation of antifungal medications in these settings were 57%, 9% and 85%, respectively. Among those screened positive for CrAg, 75% underwent lumbar puncture and 58% had cryptococcal meningitis. CrAg screening uptake and cryptococcal antigenemia was significantly lower among studies involved multiple sites compared to studies that involved single sites.

Despite of its potential benefits in reducing the development of fulminant cryptococcal meningitis and mortality associated with cryptococcal infection, compliance of the WHO guidelines in the implementation of CrAg screening among PLWH presenting with AHD is suboptimal. Previous study showed that compared to high income countries, the trajectory of cryptococcal meningitis in LMICs has not decreased despite the expansion of ART [38]. This observation may be supported by the low uptake of CrAg screening among eligible individuals leading to missed opportunity to identify people at risk of developing cryptococcal meningitis. Effective screening followed by initiation of preemptive antifungal medications for those with cryptococcal antigenemia but no evidence of cryptococcal meningitis is known to reduce the burden of cryptococcal meningitis. WHO recommended preemptive antifungal treatment for people who had positive CrAg test but without evidence of cryptococcal meningitis. While there is no standardized guidelines for managing CrAg-negative PLWH who present with AHD, the provision of fluconazole in this group of patients remains controversial [39, 40]. In this analysis, although nearly 85% of those with cryptococcal antigenemia initiated antifungal medications, nearly four in ten PLWH eligible for CrAg screening did not receive screening test. Historically, CrAg screening has been low among PLWH eligible for screening. Some of the reasons reported to be associated with the low uptake of CrAg includes unawareness of WHO guidelines, reagents stock out and dysfunctional CD4 count instruments [19–21]. Periodic training of healthcare workers and continuous medical education would increase awareness of WHO recommendations guidelines. Moreover, domestic investments in restocking and restoration of CD4 count machines need to be prioritized given the benefits of early identification and management of patients with cryptococcal antigenemia.

We observed uptake of CrAg screening to be different between studies conducted from single sites compared to those conducted from multiple sites. Those that involved participants from multiple sites had lower estimates of CrAg uptake compared to those that involved participants from single sites. Multi-site studies included health care facilities of different levels. Low level health care facilities are likely to have low level of resources and capacity to conduct CrAg screening compared to high level healthcare facilities. In this analysis, two of the four studies that were conducted from single site were conducted from tertiary hospitals that are more likely to be more resourced compared to low level healthcare facilities. The significant differences in CrAg uptake between single-site and multi-site studies underscore the need for national-level data to fully assess guideline compliance. Single-site studies, particularly those based in tertiary hospitals, may overestimate CrAg uptake due to better resource availability. In contrast, multi-site studies, which more closely represent real-world settings, highlight the challenges faced by lower-level healthcare facilities, such as stockouts of CrAg testing reagents and limited training of healthcare providers.

WHO recommendations to initiate preemptive antifungal medications to patients who screened positive for CrAg require to exclude those without evidence of cryptococcal meningitis. As part of the routine procedure towards the diagnosis of cryptococcal meningitis, patients need to undergo lumbar puncture to derive cerebral spinal fluid and tested for cryptococcal meningitis. In this analysis nearly 25% of patients who screened positive for CrAg did not undergo lumbar puncture procedure. Refusals, death prior to procedure, lack of capacity to perform the procedure could partly explain why they did not undergo lumbar puncture [19]. Since lumbar puncture is a gold standard test towards the diagnosis of cryptococcal meningitis [41], and in this analysis nearly 60% of those who underwent lumbar puncture procedure were diagnosed with cryptococcal meningitis, improving its uptake for those eligible should be enhanced.

We acknowledge the presence of limitations in this meta-analysis. This analysis included CrAg screening for data collected between 2011 and 2019. During this period, treatment guidelines changed, and some CrAg screening programs changed from provider initiated to laboratory reflex screening. The reported pooled estimate of CrAg screening uptake from this analysis may not necessarily reflect the current CrAg uptake due to these changes. Laboratory reflex CrAg screening appears to be more effective than provider initiated [23]. Moreover, an ideal CrAg screening uptake assessment would be estimated using national HIV program data that encompasses data from health care facilities of different tiers. The pooled CrAg screening uptake from this analysis used data from multiple study sites for six studies and single site from four studies. While inclusion of data from multiple sites with different levels of health care facilities provide more informative program data, the inclusion of data from single site may over or underestimate CrAg uptake depending on the level of care of the hospital/healthcare facility. CrAg screening coverage, pre-emptive antifungal treatment, and coverage of lumbar puncture might have been higher in this analysis versus what is likely occurring under routine HIV program due to unavailability of antifungal medications, CrAg screening test kits and technical ability to perform lumbar puncture procedures. More programmatic HIV data is needed to better assess these interventions. In addition, this review included both inpatient and outpatient PLWH. The prevalence of CrAg screening is likely to be higher among inpatients compared to outpatients settings. Furthermore, we restricted our search to publications written in English, it is likely that other relevant publications from non-English journals were missed. The main strength of this research is the inclusion of ten studies leadings to an overall large sample size to compute pooled estimates of CrAg screening uptake and CrAg antigenemia. This meta-analysis remains relevant as it provides estimates of the uptake of CrAg

screening to understand the magnitude of compliance of WHO recommendations in the management of PLWH presenting with AHD in Africa.

## Conclusions

In Africa, we identified significant gaps in CrAg screening among people with AHD and initiations of antifungal medications among people with positive CrAg. Our observation indicates compliance of WHO guidelines in the management of people with AHD is suboptimal. To reduce the risk of developing cryptococcal meningitis and associated mortality, national HIV programs must prioritize routine evaluations of CrAg screening uptake and ensure continuous compliance with WHO guidelines. These evaluations should include assessments of both provider-initiated and laboratory reflex-based CrAg screening to identify gaps in screening coverage, particularly in resource-limited healthcare facilities. Improved training, more reliable CD4 testing infrastructure, and greater awareness of WHO guidelines among healthcare workers are essential for achieving higher CrAg screening uptake. Efforts to improve screening for cryptococcal antigen and initiations of preemptive antifungal medications to those screened positive should be increased. We recommend routine evaluation of national data on the performance of these guidelines. These evaluations are critical to identify gaps and improve performance.

## Supporting information

**S1 Table. Prisma checklist.**
(DOCX)

**S2 Table. Systematic review search strategies.**
(DOCX)

**S3 Table. Data collection form.**
(DOCX)

**S4 Table. Summary of all studies identified through literature search.**
(DOCX)

**S5 Table. Risk of bias assessment.**
(DOCX)

**S6 Table. Overall quality of systematic review using GRADE system framework.**
(DOCX)

## Acknowledgments

We acknowledge Emilie Ludeman and Taylor Lascko for database searches and linguistic review and the authors of included studies and all the clients that contributed to their research.

## Author Contributions

**Conceptualization:** Zuhura Mbwana Ally, Habib Omari Ramadhani.

**Data curation:** Swalehe Mustafa Salim, Zarin Nudar Rodoshi, Muhidin Ibrahim Hundisa, Rebecca Mesfin Sileshi, Biruk Demisse Ayalew.

**Formal analysis:** Jackline Vicent Mbishi, Mariam Salim Mbwana, Haji Mbwana Ally.

**Investigation:** Hafidha Mhando Bakari, Lynn Moshi.

**Methodology:** Jackline Vicent Mbishi, Habib Omari Ramadhani.

**Software:** Rahma Musoke, Habib Omari Ramadhani.

**Validation:** Mariam Salim Mbwana, Hafidha Mhando Bakari, Swalehe Mustafa Salim, Zarin Nudar Rodoshi, Yousef Elias Fakhoury, Haji Mbwana Ally.

**Visualization:** Rahma Musoke.

**Writing – original draft:** Zuhura Mbwana Ally.

**Writing – review & editing:** Lynn Moshi, Yousef Elias Fakhoury, Haji Mbwana Ally, Habib Omari Ramadhani.

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
