## [Decision Letter · Decision Letter 0]

2 Sep 2024

PONE-D-24-28898Systematic review on the compliance of WHO guidelines in the management of patients with advanced HIV disease in low- and middle-income countries: The case of cryptococcal antigen screening.PLOS ONE

Dear Dr. Ally,

Thank you for submitting your manuscript to PLOS ONE. After careful consideration, we feel that it has merit but does not fully meet PLOS ONE’s publication criteria as it currently stands. Therefore, we invite you to submit a revised version of the manuscript that addresses the points raised during the review process.

We look forward to receiving your revised manuscript.

Kind regards,

Robert Jeffrey Edwards, MBBS, DrPH

Academic Editor

PLOS ONE

Journal requirements: 1. When submitting your revision, we need you to address these additional requirements. Please ensure that your manuscript meets PLOS ONE's style requirements, including those for file naming. The PLOS ONE style templates can be found at https://journals.plos.org/plosone/s/file?id=wjVg/PLOSOne_formatting_sample_main_body.pdf and https://journals.plos.org/plosone/s/file?id=ba62/PLOSOne_formatting_sample_title_authors_affiliations.pdf. 2. We suggest you thoroughly copyedit your manuscript for language usage, spelling, and grammar. If you do not know anyone who can help you do this, you may wish to consider employing a professional scientific editing service.  The American Journal Experts (AJE) (https://www.aje.com/) is one such service that has extensive experience helping authors meet PLOS guidelines and can provide language editing, translation, manuscript formatting, and figure formatting to ensure your manuscript meets our submission guidelines. Please note that having the manuscript copyedited by AJE or any other editing services does not guarantee selection for peer review or acceptance for publication.  Upon resubmission, please provide the following: The name of the colleague or the details of the professional service that edited your manuscript A copy of your manuscript showing your changes by either highlighting them or using track changes (uploaded as a *supporting information* file) A clean copy of the edited manuscript (uploaded as the new *manuscript* file)”. 3. Please include captions for your Supporting Information files at the end of your manuscript, and update any in-text citations to match accordingly. Please see our Supporting Information guidelines for more information: http://journals.plos.org/plosone/s/supporting-information.  4. As required by our policy on Data Availability, please ensure your manuscript or supplementary information includes the following:  A numbered table of all studies identified in the literature search, including those that were excluded from the analyses.   For every excluded study, the table should list the reason(s) for exclusion.   If any of the included studies are unpublished, include a link (URL) to the primary source or detailed information about how the content can be accessed.  A table of all data extracted from the primary research sources for the systematic review and/or meta-analysis. The table must include the following information for each study:  Name of data extractors and date of data extraction  Confirmation that the study was eligible to be included in the review.   All data extracted from each study for the reported systematic review and/or meta-analysis that would be needed to replicate your analyses.  If data or supporting information were obtained from another source (e.g. correspondence with the author of the original research article), please provide the source of data and dates on which the data/information were obtained by your research group.  If applicable for your analysis, a table showing the completed risk of bias and quality/certainty assessments for each study or outcome.  Please ensure this is provided for each domain or parameter assessed. For example, if you used the Cochrane risk-of-bias tool for randomized trials, provide answers to each of the signalling questions for each study. If you used GRADE to assess certainty of evidence, provide judgements about each of the quality of evidence factor. This should be provided for each outcome.   An explanation of how missing data were handled.   This information can be included in the main text, supplementary information, or relevant data repository. Please note that providing these underlying data is a requirement for publication in this journal, and if these data are not provided your manuscript might be rejected.  

Reviewers' comments:

Reviewer's Responses to Questions

**Comments to the Author**

1. Is the manuscript technically sound, and do the data support the conclusions?

Reviewer #1: Yes

Reviewer #2: Yes

Reviewer #3: Partly

Reviewer #4: Partly

2. Has the statistical analysis been performed appropriately and rigorously? 

Reviewer #1: I Don't Know

Reviewer #2: Yes

Reviewer #3: Yes

Reviewer #4: Yes

3. Have the authors made all data underlying the findings in their manuscript fully available?

Reviewer #1: Yes

Reviewer #2: Yes

Reviewer #3: Yes

Reviewer #4: No

4. Is the manuscript presented in an intelligible fashion and written in standard English?

Reviewer #1: Yes

Reviewer #2: Yes

Reviewer #3: Yes

Reviewer #4: Yes

5. Review Comments to the Author

Reviewer #1: The manuscript entitled “Systematic review on the compliance of WHO guidelines in the management of

patients with advanced HIV disease in low- and middle-income countries: The case of

cryptococcal antigen screening” by Zuhura Mbwana Ally and colleagues highlighted an important unmet need among resource-limited countries regarding the testing of cryptococcal antigen. These issues should be raised to improve HIV care in LMC. I have only minor comments in this manuscript;

1. The title should be “Systematic review on the compliance of WHO guidelines in the management of patients with advanced HIV disease in Africa: The case of cryptococcal antigen screening”, as the papers included in this study have been conducted in Africa and may not represent the whole LMC.

2. The authors should discuss or suggest about the intervention how to improve this non-compliance to WHO guideline.

3. The author should discuss what is the action if the CrAG tested to be “negative”. Is there any guideline to prescribe primary prophylaxis in those patients? There are conflict data on the use of fluconazole primary prophylaxis. Some data showed advantages (Ssekitoleko R, et al. Future Virol. 2013 Sep;8(9):10.2217/fvl.13.71. doi: 10.2217/fvl.13.71. PMID: 24368930; PMCID: PMC3869998.) but some are not (Sungkanuparph S, at al. Clin Infect Dis 2017; 64 (7), Pages 967–970). What is the practice on Africa?

Reviewer #2: The authors presents the result of a systematic analysis on CrAg screening uptake, antigenemia and preemptive treatment. Identifying gaps in CrAg screening are of major importance to reduce the overall burden of cryptococcosis in LMICs. The manuscript is well written, and the methodology is comprehensively reported. However, the choice of including studies spanning over 15 years limits the representativity of these results.

-Indeed, the biggest limitation with regards to assessing adherence to WHO guidelines for CrAg screening is that the time period covered in this analysis overlaps with different recommendations and different national screening strategies. For instance, prior to the 2018 WHO guidelines, the 2011 WHO rapid advice guidelines only stated that CrAg could be considered in patients with a CD4 counts <100 and where this population also has a high prevalence of cryptococcal antigenaemia. This is of note, as many countries have since progressively implemented screen & treat programmes. For instance, in the Vallabhaneni et al study from South Africa included in this analysis, the authors report on results of provider-initiated CrAg screening, which was shown to be relatively ineffective. Since 2017, South Africa has implemented a nationwide reflex screening programme, wherein CrAg is systematically performed on remnant blood from CD4 tests, which was shown to be much more effective than provider-initiated screening. The inclusion of these older studies therefore biases results on CrAg uptake, and means that they are not likely representative of current screening practices.

Also, please check that there are no duplicates between publications : the publications by Vallabhaneni et al, 2016 and Longley et al, 2016 have the same authors, number of patients, and results.

-One other important aspect of CrAg screening that is not mentioned is the healthcare setting in which CrAg screening is performed. This analysis includes studies in which patients are screened in various healthcare settings (various primary care settings, inpatient). Depending on the structure of the country’s healthcare setting, this may affect both the uptake of CrAg screening and the prevalence of antigenemia. Did the authors take this into account?

-Stratifying CrAg uptake on ART experience is of interest. However, results are limited by the small sample size, as highlighted by the authors, with significant heterogeneity between the two included study (sample size, definition of ART experience, inpatient vs outpatient, criteria for CrAg screening…). The results concerning CrAg screening uptake with a confidence interval of 0 to 99.5% are therefore difficult to interpret.

Minor comments:

Introduction

Line 72 and 74. More recent estimates of the global burden of cryptococcosis using 2020 data have been published, and the proportion of HIV-related deaths linked to cryptococcosis is now thought to be closer to 20% (Rajasingham et al, 2022, DOI: 10.1016/S1473-3099(22)00499-6).

Line 83. Reference 8 relates to the evaluation of newer preemptive treatment strategies. Consider replacing it one of the landmark CrAg screen & treat studies using fluconazole (Jarvis et al, 2009 DOI: 10.1086/597262, or Mfinanga et al, 2015 DOI: 10.1016/S0140-6736(15)60164-7)

Line 85. Please specify that the numbers cited are for CD4 counts <200, as the incidence of cryptococcosis among CrAg-positive individuals with CD4 counts <100 is much higher without preemptive treatment (20%, DOI: 10.1093/cid/ciy567)

Reviewer #3: This paper is well written and is aiming to address a very important topic. Our current understanding of the real-life coverage of CrAg screening and pre-emptive fluconazole treatment is quite limited, due to challenges in routine M and E and reporting from HIV programs. A systematic review of existing data on the coverage of these interventions would be a welcome addition to our understanding of the cryptococcal intervention implementation landscape.

However, this manuscript requires some major revisions before the findings can be appropriately interpreted in terms of the coverage of these interventions.

Major:

In general, it needs to be made much more clear that much of the literature being reviewed and included in the manuscript results are from single center studies and sometimes are in the context of operational research, and therefore would not be expected to necessarily reflect the actual uptake of CrAg screening and coverage of pre-emptive fluconazole under routine HIV program conditions; what is actually being reported here in the pooled results currently reflects a combination of research studies, programmatic implementation of national guidelines (but often single center, and sometimes specifically in inpatient wards only), and in some cases programmatic implementation which took place in years preceding the WHO’s inclusion of CrAg screening and pre-emptive fluconazole as part of the package of care for AHD; so the pooled coverage of CrAg screening and fluconazole taken from these studies frequently would not be necessarily expected to be reflective of real-life coverage of these interventions (and doesn’t even purport to be so in the papers themselves).

In order to accurately estimate country compliance with the relevant WHO guidance, additional data needs to be included in the review, ideally including national level, or at least multi-site, estimates of CrAg screening and pre-emptive fluconazole treatment done under more routine HIV program conditions. In recent years more data from National HIV programs and other implementing partners has become available, but not always in published literature form. For example, see AHD Dashboard: Policies for Optimal Care of Cryptococcal Meningitis - O'Neill : O'Neill (georgetown.edu).

If not possible to expand the inclusion criteria of the review analysis to capture this type of data (which is more reflective of the stated aims of your analysis), then the results of the current analysis should be re-framed to focus more on the CrAg prevalence identified (as this is an important result in of itself) and less on the coverage of CrAg screening and pre-emptive fluconazole treatment, since the coverage of these interventions in research settings or in single-center analyses are simply not generalizable to national levels.

Some of the included studies took place prior the release of the WHO advanced HIV disease package in 2017. This context should be discussed in the manuscript; additionally WHO guidance regarding the importance of screening at the 101-200 CD4 threshold and of targeting ART experienced patients was released in more recent years (2018, 2022). National country guidelines are frequently being updated to match the latest guidance and therefore the year in which the data is being pulled from is very important in terms of how an HIV program is targeting their CrAg screening program.

Line 96-97. The current cited paper from Malawi is not sufficient to make this broad statement about CrAg coverage in Malawi, as this was a single center study and was focusing only on inpatients. The Uganda reference is also older and not reflective of CrAg screening coverage in Uganda in 2024. Suggest expanding the literature review on these, or at the very least adding context to the statement regarding the limitations and years of the referenced literature.

Line 255-256. This statement is out of date, the cited paper here is from 2016 and predates a shift back to CD4 testing at baseline in many countries. Several of these countries currently recommend routine baseline CD4 for ART-naïve as well as in the case of ART interruption/reengagement with care and in cases of suspected/confirmed ART failure; I recommend that the author reviews most recent national HIV program guidelines (which are available online).

Minor:

Lines 84-85 and 90-91. It should be made clear that for patients with a positive blood CrAg result, it is crucial to rule out concurrent cryptococcal meningitis. This should be made clear so that the reader does not interpret this as meaning that all blood CrAg positive patients need pre-emptive fluconazole only.

In general in the results, it should be made more clear how many countries are covered in the review.

Line 75. I suggest updating this to the most recent global estimates (Rajasingham et al 2022).

Line 88. Suggest expanding the review of literature on this important contextual piece of information. Could consider including papers from Temfack et al and Jarvis et al.

Line 273-274, this appears to say that VL results are to be used as a criteria for CrAg screening, which is not the case.

Reviewer #4: Major comments:

1. More context is needed for these studies, particularly around the populations, settings, and country guidelines at the time of screening. A study of inpatients where screening was only conducted on those with CM symptoms (e.g. Mbewe at al 2022) should not be directly compared to a study of outpatient screening at a set CD4 count range without this context provided. This will bias towards higher uptake, as testing was performed in a diagnostic rather than routine screening capacity, and will also bias towards higher CrAg positivity and treatment initiation. Also each study should be briefly described in the context in which it was conducted (i.e. hospital/clinic, inpatient/outpatient, routine programmatic assessment vs AHD pilot or focused implementation at select sites, country guidelines and recommendations in place at time of study).

2. The analysis needs to parse apart studies focused on CM diagnosis and/or workup vs outpatient screening programs. Also referral and/or CM workout should be explicitly included rather than only pre-emptive antifungal initiation as an outcome, given that for those with CM symptoms, pre-emptive fluconazole therapy would not be the outcome of choice based on guidelines at the time of some of these studies. Rather, referral for lumbar puncture and CM workup would be.

Minor Comments:

1. References for the studies included in each category should be provided in each part of tables pooling numbers.

2. Table 1:"type of population" should be specified as "ART status at time of screening". Also some of these studies are mixed but not denoted correctly on this table (e.g. Mbewe et al 2022 indicated as "experienced" when 8% of those in the study were ART-naive). Further clarity on this table is also needed in the text. It is not immediately clear why the Baluku 2021 study is listed twice, separating ART-naive and experienced. Similar with the Heller 2022 paper. These studies should be explained and given greater context as to why they were analyzed separately, and a number assigned to the sub-studies used for reference in the Table 2 calculations. Otherwise it is unclear how the Table 2 calculations were derived, from which studies, and which subsets within studies.

3. Eligibility criteria: CrAg eligibility is currently separated into CD4 < 100, CD4 < 200. Some guidelines recommend CD4 ≤200 or CD4 ≤100. Minor difference, but if slight changes were made to fit these studies in categories, this should be explicitly mentioned.

4. Table 2: There are 17 studies listed, but only 16 studies listed in Table 1. It is not clear on the table which studies are included where. References should be provided on the table for clarity. Also under CrAg antigenemia, it does not make sense that the pooled prevalence in the CD4 < 200 category is 9.9 and higher than the CD4 <100 and CD4 100-200 categories. Generally if referring to screening, it would be somewhere in between these 2 numbers. This indicates missing context, likely with a bias towards inpatient CM workup. This needs to be explicitly stated and context provided. Also the "CD4 < 100-200" category needs to be revised to "CD4 100-200" (I assume this is the intended category), or better explained.

5. Study periods indicated in Table 1 should be confirmed. Example: Blankley et 2019 study included patients from 2007-2016, not 2015-2016.

6. Would recommend using "CrAg screening uptake" or "coverage" rather than "CrAg screening prevalence", as this is easily confused with "CrAg prevalence" and the difference not well described in the paper.

7. In eligibility criteria section, should explicitly define people with advanced HIV disease. Needs clarity as to whether only CD4 count was used as a threshold or if WHO staging was also used. In some of the studies indicated, WHO staging was also included.

6. PLOS authors have the option to publish the peer review history of their article (what does this mean?). If published, this will include your full peer review and any attached files.

Reviewer #1: No

Reviewer #2: No

Reviewer #3: No

Reviewer #4: **Yes: **Greg Greene

---

## [Author Response · Author response to Decision Letter 0]

2 Oct 2024

Dear editors PLOS ONE journal,

This letter serves as response to reviewer’s comments for the manuscript “Systematic review on the compliance of WHO guidelines in the management of patients with advanced HIV disease in Africa: The case of cryptococcal antigen screening.” We greatly appreciate editors and reviewers and believe their comments have significantly improved our manuscript.

Below are point by point responses

Reviewer #1: The manuscript entitled “Systematic review on the compliance of WHO guidelines in the management of

patients with advanced HIV disease in low- and middle-income countries: The case of

cryptococcal antigen screening” by Zuhura Mbwana Ally and colleagues highlighted an important unmet need among resource-limited countries regarding the testing of cryptococcal antigen. These issues should be raised to improve HIV care in LMC. I have only minor comments in this manuscript;

1. The title should be “Systematic review on the compliance of WHO guidelines in the management of patients with advanced HIV disease in Africa: The case of cryptococcal antigen screening”, as the papers included in this study have been conducted in Africa and may not represent the whole LMC.

We thank you for your comments. We agree majority of the papers included in this study were from Africa, however, the search was for LMC and there was a paper from Taiwan. We excluded the paper from Taiwan because it included patients before WHO guidelines were implemented and edited the title accordingly.

2. The authors should discuss or suggest about the intervention how to improve this non-compliance to WHO guideline.

We thank you for your comments. Reasons for low uptake were discussed and cited. We have provided recommendations to improve CrAG uptake.

3. The author should discuss what is the action if the CrAG tested to be “negative”. Is there any guideline to prescribe primary prophylaxis in those patients? There are conflict data on the use of fluconazole primary prophylaxis. Some data showed advantages (Ssekitoleko R, et al. Future Virol. 2013 Sep;8(9):10.2217/fvl.13.71. doi: 10.2217/fvl.13.71. PMID: 24368930; PMCID: PMC3869998.) but some are not (Sungkanuparph S, at al. Clin Infect Dis 2017; 64 (7), Pages 967–970). What is the practice on Africa?

We thank you for your comments. No recommendations for those with negative CrAg screen test. We think practices is at the discretion of the provider.

Reviewer #2: The authors presents the result of a systematic analysis on CrAg screening uptake, antigenemia and preemptive treatment. Identifying gaps in CrAg screening are of major importance to reduce the overall burden of cryptococcosis in LMICs. The manuscript is well written, and the methodology is comprehensively reported. However, the choice of including studies spanning over 15 years limits the representativity of these results.

We thank you for your comments. We have modified our inclusion criteria to include studies from 2011 onwards. This is the time when WHO guidelines on management of advanced HIV disease were launched. We have included this observation as a limitation.

-Indeed, the biggest limitation with regards to assessing adherence to WHO guidelines for CrAg screening is that the time period covered in this analysis overlaps with different recommendations and different national screening strategies. For instance, prior to the 2018 WHO guidelines, the 2011 WHO rapid advice guidelines only stated that CrAg could be considered in patients with a CD4 counts <100 and where this population also has a high prevalence of cryptococcal antigenaemia. This is of note, as many countries have since progressively implemented screen & treat programmes. For instance, in the Vallabhaneni et al study from South Africa included in this analysis, the authors report on results of provider-initiated CrAg screening, which was shown to be relatively ineffective. Since 2017, South Africa has implemented a nationwide reflex screening programme, wherein CrAg is systematically performed on remnant blood from CD4 tests, which was shown to be much more effective than provider-initiated screening. The inclusion of these older studies therefore biases results on CrAg uptake, and means that they are not likely representative of current screening practices.

We thank you for your comments. These are critical observations. We have discussed these limitations. It would be great to stratify the uptake of CrAg by provider initiated and reflex testing but there are not enough studies published on reflex testing that reported CrAg uptake. We acknowledge the changes on guidelines over time.

Also, please check that there are no duplicates between publications : the publications by Vallabhaneni et al, 2016 and Longley et al, 2016 have the same authors, number of patients, and results.

We thank you for your comments. Although first author names were different, these publications are indeed duplicates. We have deleted one of them.

-One other important aspect of CrAg screening that is not mentioned is the healthcare setting in which CrAg screening is performed. This analysis includes studies in which patients are screened in various healthcare settings (various primary care settings, inpatient). Depending on the structure of the country’s healthcare setting, this may affect both the uptake of CrAg screening and the prevalence of antigenemia. Did the authors take this into account?

We thank you for your comments, we have added the context from which these studies were done and discussed possible variations in CrAG uptake due to the different levels of healthcare settings.

-Stratifying CrAg uptake on ART experience is of interest. However, results are limited by the small sample size, as highlighted by the authors, with significant heterogeneity between the two included study (sample size, definition of ART experience, inpatient vs outpatient, criteria for CrAg screening…). The results concerning CrAg screening uptake with a confidence interval of 0 to 99.5% are therefore difficult to interpret.

We thank you for your comments. We decided not to report stratified analysis based on the status of ART at the time of screening due to limited number of studies that reported uptake among ART experienced. 

Minor comments:

Introduction

Line 72 and 74. More recent estimates of the global burden of cryptococcosis using 2020 data have been published, and the proportion of HIV-related deaths linked to cryptococcosis is now thought to be closer to 20% (Rajasingham et al, 2022, DOI: 10.1016/S1473-3099(22)00499-6).

We thank you for your comments. We have used this reference as recommended. 

Line 83. Reference 8 relates to the evaluation of newer preemptive treatment strategies. Consider replacing it one of the landmark CrAg screen & treat studies using fluconazole (Jarvis et al, 2009 DOI: 10.1086/597262, or Mfinanga et al, 2015 DOI: 10.1016/S0140-6736(15)60164-7)

We thank you for your comments. We have added Jarvis et al reference as recommended.

Line 85. Please specify that the numbers cited are for CD4 counts <200, as the incidence of cryptococcosis among CrAg-positive individuals with CD4 counts <100 is much higher without preemptive treatment (20%, DOI: 10.1093/cid/ciy567)

We thank you for your comments. We have specified CD4 count for that reference as advised. 

Reviewer #3: This paper is well written and is aiming to address a very important topic. Our current understanding of the real-life coverage of CrAg screening and pre-emptive fluconazole treatment is quite limited, due to challenges in routine M and E and reporting from HIV programs. A systematic review of existing data on the coverage of these interventions would be a welcome addition to our understanding of the cryptococcal intervention implementation landscape.

However, this manuscript requires some major revisions before the findings can be appropriately interpreted in terms of the coverage of these interventions.

Major:

In general, it needs to be made much more clear that much of the literature being reviewed and included in the manuscript results are from single center studies and sometimes are in the context of operational research, and therefore would not be expected to necessarily reflect the actual uptake of CrAg screening and coverage of pre-emptive fluconazole under routine HIV program conditions; what is actually being reported here in the pooled results currently reflects a combination of research studies, programmatic implementation of national guidelines (but often single center, and sometimes specifically in inpatient wards only), and in some cases programmatic implementation which took place in years preceding the WHO’s inclusion of CrAg screening and pre-emptive fluconazole as part of the package of care for AHD; so the pooled coverage of CrAg screening and fluconazole taken from these studies frequently would not be necessarily expected to be reflective of real-life coverage of these interventions (and doesn’t even purport to be so in the papers themselves).

We thank you for your comments. We agree variations in the type of studies used to provide pooled estimates of CrAg uptake. We have added a table with context from which studies were conducted. We have also deleted studies that reported CrAg coverage before WHO guidelines were in place. As previously noted by the reviewer, the lack of routine monitoring and evaluation limits data availability. We have discussed limitations of our pooled estimates.

In order to accurately estimate country compliance with the relevant WHO guidance, additional data needs to be included in the review, ideally including national level, or at least multi-site, estimates of CrAg screening and pre-emptive fluconazole treatment done under more routine HIV program conditions. In recent years more data from National HIV programs and other implementing partners has become available, but not always in published literature form. For example, see AHD Dashboard: Policies for Optimal Care of Cryptococcal Meningitis - O'Neill : O'Neill (georgetown.edu).

If not possible to expand the inclusion criteria of the review analysis to capture this type of data (which is more reflective of the stated aims of your analysis), then the results of the current analysis should be re-framed to focus more on the CrAg prevalence identified (as this is an important result in of itself) and less on the coverage of CrAg screening and pre-emptive fluconazole treatment, since the coverage of these interventions in research settings or in single-center analyses are simply not generalizable to national levels.

We thank you for your comments. These are very important observations. Of the 10 studies included now, 4 studies reported data from single sites. All other 6 studies were multi sites or used data from the National databases repositories. We have added this information in the manuscript. The use of National databases with more routine HIV program conditions would be ideal for the estimation of pooled CrAG uptake, however, as pointed out by the reviewer, the data is not available in the literature, limiting our ability to include them. We have stratified pooled estimates of CrAG uptake from single site studies and multi center studies to explore the differences.

Some of the included studies took place prior the release of the WHO advanced HIV disease package in 2017. This context should be discussed in the manuscript; additionally WHO guidance regarding the importance of screening at the 101-200 CD4 threshold and of targeting ART experienced patients was released in more recent years (2018, 2022). National country guidelines are frequently being updated to match the latest guidance and therefore the year in which the data is being pulled from is very important in terms of how an HIV program is targeting their CrAg screening program.

We thank you for your comments. We appreciate your important observations. We have discussed changes of the guidelines over time which may have impact on the uptake of CrAg screening. We have also deleted CrAg uptake data among ART experienced due to lack of enough data.

Line 96-97. The current cited paper from Malawi is not sufficient to make this broad statement about CrAg coverage in Malawi, as this was a single center study and was focusing only on inpatients. The Uganda reference is also older and not reflective of CrAg screening coverage in Uganda in 2024. Suggest expanding the literature review on these, or at the very least adding context to the statement regarding the limitations and years of the referenced literature.

We thank you for your comments. We have provided context to these statements. Would be ideal to expand literature on CrAg uptake, however, most of the studies done reported prevalence of CrAg positivity and not necessarily uptake among those eligible for CrAg screening.

Line 255-256. This statement is out of date, the cited paper here is from 2016 and predates a shift back to CD4 testing at baseline in many countries. Several of these countries currently recommend routine baseline CD4 for ART-naïve as well as in the case of ART interruption/reengagement with care and in cases of suspected/confirmed ART failure; I recommend that the author reviews most recent national HIV program guidelines (which are available online).

We thank you for your comments. We have edited this section.

Minor:

Lines 84-85 and 90-91. It should be made clear that for patients with a positive blood CrAg result, it is crucial to rule out concurrent cryptococcal meningitis. This should be made clear so that the reader does not interpret this as meaning that all blood CrAg positive patients need pre-emptive fluconazole only.

We thank you for your comments. We have taken note and made changes.

In general in the results, it should be made more clear how many countries are covered in the review.

We thank you for your comments. We have added this information in the results section

Line 75. I suggest updating this to the most recent global estimates (Rajasingham et al 2022).

We thank you for your comments. We used this most recent reference.

Line 88. Suggest expanding the review of literature on this important contextual piece of information. Could consider including papers from Temfack et al and Jarvis et al.

We thank you for your comments: It was also suggested by other reviewer to include Jarvis as reference around this section and has been done as recommended.

Line 273-274, this appears to say that VL results are to be used as a criteria for CrAg screening, which is not the case.

We thank you for your comments: We had edited this section.

Reviewer #4: Major comments:

1. More context is needed for these studies, particularly around the populations, settings, and country guidelines at the time of screening. A study of inpatients where screening was only conducted on those with CM symptoms (e.g. Mbewe at al 2022) should not be directly compared to a study of outpatient screening at a set CD4 count range without this context provided. This will bias towards higher uptake, as testing was performed in a diagnostic rather than routine screening capacity, and will also bias towards higher CrAg positivity and treatment initiation. Also each study should be briefly described in the context in which it was conducted (i.e. hospital/clinic, inpatient/outpatient, routine programmatic assessment vs AHD pilot or focused implementation at select sites, country guidelines and recommendations in place at time of study).

We thank you for your comments. We have provided a new table with detailed descriptions of setting, number of sites, country guidelines, routine programmatic assessment, design and data sources.

2. The analysis needs to parse apart studies focused on CM diagnosis and/or workup vs outpatient screening programs. Also referral and/or CM workout should be explicitly included rather than only pre-emptive antifungal initiation as an outcome, given that for those with CM symptoms, pre-emptive fluconazole therapy would not be the outcome of choice based on guidelines at the time of some of these studies. Rather, referral for lumbar puncture and CM workup would be.

We thank you for your comments, we have added CM work out among those with posit

---

## [Decision Letter · Decision Letter 1]

21 Oct 2024

PONE-D-24-28898R1Systematic review on the compliance of WHO guidelines in the management of patients with advanced HIV disease in Africa: The case of cryptococcal antigen screening.PLOS ONE

Dear Dr. Ally,

Thank you for submitting your manuscript to PLOS ONE. After careful consideration, we feel that it has merit but does not fully meet PLOS ONE’s publication criteria as it currently stands. Therefore, we invite you to submit a revised version of the manuscript that addresses the points raised during the review process.

We look forward to receiving your revised manuscript.

Kind regards,

Robert Jeffrey Edwards, MBBS, DrPH

Academic Editor

PLOS ONE

Journal Requirements:

Additional Editor Comments:

Please revise the manuscript in accordance with the revisions suggested by the reviewer

Reviewers' comments:

Reviewer's Responses to Questions

**Comments to the Author**

1. If the authors have adequately addressed your comments raised in a previous round of review and you feel that this manuscript is now acceptable for publication, you may indicate that here to bypass the “Comments to the Author” section, enter your conflict of interest statement in the “Confidential to Editor” section, and submit your "Accept" recommendation.

Reviewer #1: All comments have been addressed

Reviewer #3: (No Response)

Reviewer #5: All comments have been addressed

2. Is the manuscript technically sound, and do the data support the conclusions?

Reviewer #1: Yes

Reviewer #3: Yes

Reviewer #5: Yes

3. Has the statistical analysis been performed appropriately and rigorously? 

Reviewer #1: Yes

Reviewer #3: Yes

Reviewer #5: Yes

4. Have the authors made all data underlying the findings in their manuscript fully available?

Reviewer #1: Yes

Reviewer #3: Yes

Reviewer #5: Yes

5. Is the manuscript presented in an intelligible fashion and written in standard English?

Reviewer #1: Yes

Reviewer #3: Yes

Reviewer #5: Yes

6. Review Comments to the Author

Reviewer #1: This is a revised manuscript I reviewed before. From the reviewer's response to my comments and other comments, the authors addressed the reviewer's comments point-by-point and now it is suitable for publication.

Reviewer #3: Comments:

Line 28-29: this statement on CrAg negative seems to be a bit less relevant to the focus of the paper

line 31: Please make it clear that the estimates you derived are from available selected published studies. Stating that the stimates are for "Africa" gives the impression that the estimates in this analysis are much more comprehensive than they are.

Line 44-45: Suggest making it clear that the prevalence of antigenemia is what was found to be lower in the multi-site studies versus single sites

Lines 90-91: Suggest making it clear that this line is referring to a different study (ref #11) and maybe also commenting briefly on the differences between the ref #10 and ref #11in terms of included studies and locations

lines 99-100: Please avoid making statements that imply a national estimate when referencing studies/analyses which had only a few sites (or only one).

line 100-101: It appears that this statement on improved coverage in ART naive patients in Uganda may be referencing a study that was analysing ART-experienced patients with viral unsuppression; please review and edit as needed

Lines 307-310: Somewhere in the limitations section, I suggest commenting even more on the reasons why the estimates of CrAg screening coverage, pre-emptive treatment, and coverage of LP might have been higher in this analysis versus what is likely occurring under routine HIV program settings (i.e. in a setting where study is being conducted things like LPs, fluconazole, CrAg kits, etc are more likely to be avialable) and why the true coverage may be lower in reality. Could use this to mention the need for more programmatic HIV program data on these interventions.

I also suggest adding more mention of the combination of inpatient and outpatient settings included in the reviewed papers (prevalence of CrAg typically going to be higher in HIV+ inpatients versus outpatients).

Line 354-355: Lumbar puncture itself is not a test, it’s the procedure needed to derive the CSF specimen which would then be tested for cryptococcus.

Reviewer #5: The authors have responded adequately to the comments. Therefore, this revised manuscript is suitable for publication.

7. PLOS authors have the option to publish the peer review history of their article (what does this mean?). If published, this will include your full peer review and any attached files.

Reviewer #1: **Yes: **Methee Chayakulkeeree

Reviewer #3: No

Reviewer #5: **Yes: **Methee Chayakulkeeree

---

## [Author Response · Author response to Decision Letter 1]

22 Oct 2024

Dear editors PLOS ONE journal,

This letter serves as response to reviewer’s comments for the manuscript “Systematic review on the compliance of WHO guidelines in the management of patients with advanced HIV disease in Africa: The case of cryptococcal antigen screening.” We greatly appreciate editors and reviewers and believe their comments have significantly improved our manuscript.

Below are point by point responses

Reviewer #1: This is a revised manuscript I reviewed before. From the reviewer's response to my comments and other comments, the authors addressed the reviewer's comments point-by-point and now it is suitable for publication.

We are grateful we managed to respond to all your earlier comments.

Reviewer #3: Comments:

Line 28-29: this statement on CrAg negative seems to be a bit less relevant to the focus of the paper.

We thank you for your comments. The statement has been revised.

line 31: Please make it clear that the estimates you derived are from available selected published studies. Stating that the stimates are for "Africa" gives the impression that the estimates in this analysis are much more comprehensive than they are.

We thank you for your comments. The statement has been revised.

Line 44-45: Suggest making it clear that the prevalence of antigenemia is what was found to be lower in the multi-site studies versus single sites

We thank you for your comments. Both CrAg uptake and the prevalence of antigenemia were lower in studies that involved multiple sites vs single sites. Line 44-45 refers to CrAg uptake comparisons (multi vs single site) while line 46-48 refers to prevalence of antigenemia comparisons (multi vs single site).No changes were made.

Lines 90-91: Suggest making it clear that this line is referring to a different study (ref #11) and maybe also commenting briefly on the differences between the ref #10 and ref #11in terms of included studies and locations.

We thank you for your comments. We have provided some additional details about these two references.

lines 99-100: Please avoid making statements that imply a national estimate when referencing studies/analyses which had only a few sites (or only one).

We thank you for your comments. We made changes where these statements were made.

line 100-101: It appears that this statement on improved coverage in ART naive patients in Uganda may be referencing a study that was analysing ART-experienced patients with viral unsuppression; please review and edit as needed

We thank you for your comments. We have edited this section.

Lines 307-310: Somewhere in the limitations section, I suggest commenting even more on the reasons why the estimates of CrAg screening coverage, pre-emptive treatment, and coverage of LP might have been higher in this analysis versus what is likely occurring under routine HIV program settings (i.e. in a setting where study is being conducted things like LPs, fluconazole, CrAg kits, etc are more likely to be avialable) and why the true coverage may be lower in reality. Could use this to mention the need for more programmatic HIV program data on these interventions.

I also suggest adding more mention of the combination of inpatient and outpatient settings included in the reviewed papers (prevalence of CrAg typically going to be higher in HIV+ inpatients versus outpatients).

We thank you for your comments. We have incorporated these suggestions in the limitation section.

Line 354-355: Lumbar puncture itself is not a test, it’s the procedure needed to derive the CSF specimen which would then be tested for cryptococcus.

We thank you for your comments. We have edited this section

Reviewer #5: The authors have responded adequately to the comments. Therefore, this revised manuscript is suitable for publication.

We are grateful we managed to respond to all your earlier comments.

---

## [Editor Report · Decision Letter 2]

24 Oct 2024

Systematic review on the compliance of WHO guidelines in the management of patients with advanced HIV disease in Africa: The case of cryptococcal antigen screening.

PONE-D-24-28898R2

Dear Dr. Mbwana Ally

We’re pleased to inform you that your manuscript has been judged scientifically suitable for publication and will be formally accepted for publication once it meets all outstanding technical requirements.

Kind regards,

Robert Jeffrey Edwards, MBBS, DrPH

Academic Editor

PLOS ONE

Additional Editor Comments (optional):

The comments by the reviewers have been addressed
---

## [Editor Report · Acceptance letter]

15 Dec 2024

PONE-D-24-28898R2 

PLOS ONE

Dear Dr. Ally, 

I'm pleased to inform you that your manuscript has been deemed suitable for publication in PLOS ONE. Congratulations! Your manuscript is now being handed over to our production team.

Kind regards, 

on behalf of

Dr. Robert Jeffrey Edwards 

Academic Editor

PLOS ONE